# Evolution-guided functional analyses reveal diverse antiviral specificities encoded by IFIT1 genes in mammals

Matthew D Daugherty[1,2*†], Aaron M Schaller[1‡], Adam P Geballe[3,4,5,6], Harmit S Malik[1,2*]

[1]Division of Basic Sciences, Fred Hutchinson Cancer Research Center, Seattle, United States; [2]Howard Hughes Medical Institute, Fred Hutchinson Cancer Research Center, Seattle, United States; [3]Divisions of Human Biology, Fred Hutchinson Cancer Research Center, Seattle, United States; [4]Clinical Research, Fred Hutchinson Cancer Research Center, Seattle, United States; [5]Department of Microbiology, University of Washington School of Medicine, Seattle, United States; [6]Department of Medicine, University of Washington School of Medicine, Seattle, United States

*For correspondence:
mddaugherty@ucsd.edu (MDD);
hsmalik@fhcrc.org (HSM)

Present address: †Molecular Biology Section, Division of Biological Sciences, University of California, San Diego, San Diego, United States; ‡Molecular and Cellular Biology Graduate Program, University of California, Berkeley, Berkeley, United States

Competing interests: The authors declare that no competing interests exist.

**Abstract** IFIT (interferon-induced with tetratricopeptide repeats) proteins are critical mediators of mammalian innate antiviral immunity. Mouse IFIT1 selectively inhibits viruses that lack 2'O-methylation of their mRNA 5' caps. Surprisingly, human IFIT1 does not share this antiviral specificity. Here, we resolve this discrepancy by demonstrating that human and mouse IFIT1 have evolved distinct functions using a combination of evolutionary, genetic and virological analyses. First, we show that human *IFIT1* and mouse *IFIT1* (renamed *IFIT1B*) are not orthologs, but are paralogs that diverged >100 mya. Second, using a yeast genetic assay, we show that IFIT1 and IFIT1B proteins differ in their ability to be suppressed by a cap 2'O-methyltransferase. Finally, we demonstrate that IFIT1 and IFIT1B have divergent antiviral specificities, including the discovery that only IFIT1 proteins inhibit a virus encoding a cap 2'O-methyltransferase. These functional data, combined with widespread turnover of mammalian *IFIT* genes, reveal dramatic species-specific differences in IFIT-mediated antiviral repertoires.

## Introduction

Mammalian antiviral defenses rely on the combined functions of a vast collection of immunity genes. One important arm of the immune system is innate, or cell intrinsic, antiviral immunity, which establishes an antiviral state in cells by signaling through the cytokine interferon (IFN) and subsequent upregulation of hundreds of genes known collectively as IFN-stimulated genes (ISGs). ISGs serve as the first line of defense against viruses and typically function to sense pathogen-associated molecular patterns (PAMPs) or directly restrict virus replication (*Schneider et al., 2014*; *Schoggins, 2014*).

Among the most highly upregulated ISGs are members of the *IFIT* (interferon-induced with tetratricopeptide repeats) genes. Upon IFN-stimulation or viral infection, the mRNA levels of *IFITs* increase 100- to 1000-fold, and IFIT proteins have been implicated in inhibition of a broad range of viruses (*Diamond and Farzan, 2013*; *Fensterl and Sen, 2015*; *Vladimer et al., 2014*). However, the number and identity of *IFIT* genes can vary substantially between species. For instance, while humans have five intact *IFIT* genes (*IFIT1, 1B, 2, 3* and *5*), rats have four (*IFIT1, 1b, 2* and *3*) and mice have six (*IFIT1, 1b, 1c, 2, 3* and *3b*) (*Fensterl and Sen, 2011*; *2015*; *Liu et al., 2013*). The functional consequences of *IFIT* family evolution are unknown, in part because the antiviral functions and specificities of IFITs are incompletely characterized. Initial studies with IFIT1 and IFIT2 from humans and

**eLife digest** When a virus is detected in the body, hundreds of different proteins in the immune system are rapidly produced as a first line of defense to limit the ability of the virus to multiply and spread. Many of these 'innate' immunity proteins have rapidly evolved in mammals in escalating molecular 'arms races' with the viruses they target. This makes it more difficult to work out exactly what job each protein performs. Even when the role of a specific protein has been determined in mice, for example, it does not always follow that the human protein with the same name will perform the same role.

The IFIT1 proteins are some of the most highly produced innate immunity proteins in mammals during viral infections. In the infected cell, host and viral proteins are both made from templates made of molecules of ribonucleic acid (RNA). Previous work showed that the IFIT1 protein in mice is able to exploit a critical chemical difference between host and virus RNA to selectively block the production of virus proteins. However, other research suggests that the human IFIT1 protein does not use the same chemical difference to distinguish between host and virus RNA.

Here, Daugherty et al. unravel the complicated evolutionary history of IFIT1 proteins to show that the mouse and human proteins are not as closely related to each other as first thought. Instead, they belong to two different protein families with distinct roles in fighting viruses. Further experiments show that the human and mouse IFIT proteins likely discriminate between host and viral RNA using different cues, leading to their action against different sets of viruses.

Daugherty et al.'s findings highlight that there are additional undiscovered chemical differences between host and viral RNA that the immune system can exploit to selectively target and stop viruses from multiplying. Furthermore, these findings re-emphasize the often-overlooked differences between the immune systems of mice and humans. The finding that mammals have such a diverse set of IFIT1 immunity proteins may directly explain why different species are susceptible to different viruses.

---

mice indicated that these proteins might mediate their antiviral activity by inhibiting mRNA translation through interactions with the translation initiation factor eIF3 (*Guo et al., 2000*; *Hui et al., 2003*; *Terenzi et al., 2005*). In this way, IFITs appeared to function similarly to another critical mediator of the innate immune system, Protein Kinase R (PKR), by globally inhibiting mRNA translation. In the case of PKR, the recognition of cytoplasmic double-stranded RNA, a by-product of viral replication, triggers its activity and the global repression of protein synthesis (*Dever et al., 2007*). Such a 'self versus non-self' molecular pattern has been more enigmatic for IFIT proteins, and it has been challenging to determine how IFITs discriminate viral from host RNAs to repress viral replication specifically.

An elegant means by which one IFIT protein distinguishes 'self versus non-self' mRNAs was revealed by recent studies on mouse IFIT1. During mammalian mRNA processing, the 5' cap region is 2'O-methylated from a cap0-structure (7mGpppN, where 7mG is the 7-methyl guanosine, ppp is the triphosphate linkage, and N is any nucleotide) to a cap1-structure (7mGpppNm) (*Banerjee, 1980*). This reaction is catalyzed in the host nucleus by a dedicated 2'O-methyltransferase, known as a cap1-methyltransferase (hCMTR1 in humans) (*Belanger et al., 2010*). Interestingly, many viruses have evolved ways to mimic host cap1-structures (*Banerjee, 1980*; *Decroly et al., 2012*). For several viruses that replicate in the cytoplasm, such as poxviruses, flaviviruses, coronaviruses, and rhabdoviruses, 2'O-methylation of the cap is catalyzed by a virally-encoded cap1-methyltransferase. For other viruses, such as orthomyxoviruses, arenaviruses, and bunyaviruses, the effect is achieved by 'cap-snatching', in which a segment of cap1-modified host mRNA is appended to viral mRNAs. Either strategy results in methylated (cap1-) mRNAs, suggesting that unmethylated (cap0-) mRNAs could be recognized as a 'non-self' pattern that elicits host immunity. Indeed, mouse IFIT1 was discovered to inhibit replication of numerous viruses naturally lacking or mutated to lack 2'O-methylation by directly binding and inhibiting translation of cap0-mRNAs (*Daffis et al., 2010*; *Hyde et al., 2014*; *Ma et al., 2014*; *Menachery et al., 2014*; *Szretter et al., 2012*; *Zust et al., 2011*; *Habjan et al., 2013*; *Kimura et al., 2013*; *Kumar et al., 2014*). In this way, mouse IFIT1 selectively

inhibits viruses that translate proteins from 'non-self' cap0-mRNAs, while the host protects itself via 'self' cap1-structures on its mRNAs (*Diamond, 2014*; *Hyde and Diamond, 2015*).

Given the importance of the cap0-mRNA versus cap1-mRNA specificity in directing mouse IFIT1's repressive effects against viruses, we might expect that other mammalian *IFIT1* genes would preserve such discrimination. Surprisingly, studies on human IFIT1 have belied this expectation. For instance, human IFIT1 was shown to inhibit mRNA translation and replication of parainfluenza virus 5 (PIV5), despite the fact that PIV5 encodes a cap1-methyltransferase and PIV5 mRNAs are 2'O-methylated on their caps (*Andrejeva et al., 2013*). Other studies have implicated human IFIT1 in inhibition of hepatitis C virus (HCV) (*Raychoudhuri et al., 2011*; *Wang et al., 2003*), human papillomavirus (HPV) (*Terenzi et al., 2008*), influenza A virus (IAV) and vesicular stomatitis virus (VSV) (*Pichlmair et al., 2011*), none of which are predicted to translate proteins from cap0-mRNAs.

These seemingly contradictory results regarding the antiviral specificities of mouse IFIT1 and human IFIT1 have led to a conundrum in the field regarding the molecular functions and antiviral specificity of IFIT proteins in general. However, one implicit assumption underlying the expectation that human and mouse IFIT1 should function similarly is that mouse and human *IFIT1* represent orthologous genes. Here, we show that this is not the case. Using detailed phylogenetic analyses of *IFIT* genes in vertebrates, made possible by deconvolving the confounding effects of recurrent gene conversion, we show that human *IFIT1* and mouse *IFIT1* are two distinct paralogous genes that diverged early in mammalian evolution. Mouse genomes only have representatives of one of these paralogous genes (which we rename *IFIT1B*), whereas several primates have lost *IFITB* but retained *IFIT1*. We further show that these two divergent *IFIT1* paralogs have distinct specificities. Using a genetic assay we developed in budding yeast (which exploits yeast's inherent lack of a cap1-methyltransferase), we show that mouse and other IFIT1B proteins, but not IFIT1 proteins, discriminate cap0- from cap1-mRNA methylation by selectively inhibiting growth only when a cap1-methyltransferase is missing. Consistent with this activity, we show that only IFIT1B proteins can inhibit replication of a virus that lacks functional cap1-methylation. In contrast, we find that human and other IFIT1 proteins can inhibit the growth of yeast or of a virus that encodes a cap1-methyltransferase. Our findings thus resolve the apparent mystery of IFIT1's altered antiviral discrimination mechanism in mouse and human and delineate an important role for human IFIT1 in restriction of viruses including those that produce cap1-mRNAs. Our analyses also reveal a high degree of dynamism and alternate retention of *IFIT1* or *IFIT1B* paralogs in mammalian genomes. This *IFIT* gene turnover might profoundly impact the spectrum of viruses that different mammalian species are capable of restricting.

## Results

### Mouse IFIT1 and Human IFIT1 are not orthologous but belong to distinct gene families

The expectation that mouse and human *IFIT1* should have a similar antiviral function and specificity implicitly assumes that these two genes are orthologous, *i.e.,* they diverged when the common ancestors of humans and mice diverged as species. However, there is evidence suggesting that this assumption may not be valid. First, there have been extensive changes to the number and identity of *IFIT* genes between humans and mice ([*Fensterl and Sen, 2011*;*2015*; *Liu et al., 2013*] and *Figure 1A*). Furthermore, mouse IFIT1 is more similar at the sequence level to another human IFIT, the poorly characterized human IFIT1B, than to human IFIT1 (57% versus 53% pairwise amino acid identity). Based on these data, and the contradictory functional data for mouse IFIT1 and human IFIT1 antiviral activities (*Andrejeva et al., 2013*; *Daffis et al., 2010*; *Pichlmair et al., 2011*; *Pinto et al., 2015*), we formally tested for *IFIT1* orthology between mouse and human. To do so, we first undertook detailed phylogenetic analyses of the *IFIT* genes found in several well-assembled mammalian genomes (*Figure 1A*).

Our phylogenetic analyses were facilitated by the fact that *IFIT* genes are located in a single locus in most mammalian species (*Liu et al., 2013*). For example, human chromosome 10 contains a single locus that encodes all five intact IFIT genes: *IFIT1*, *IFIT1B*, *IFIT2*, *IFIT3* and *IFIT5* (*Figure 1A*). We found the *IFIT* locus organization is similar in many other mammalian genomes including African green monkeys, rabbits, cats, ferrets, and armadillos (*Figure 1A*). In this shared syntenic arrangement, two *IFIT1* paralogs are found next to each other in the same sense orientation as *IFIT2* and

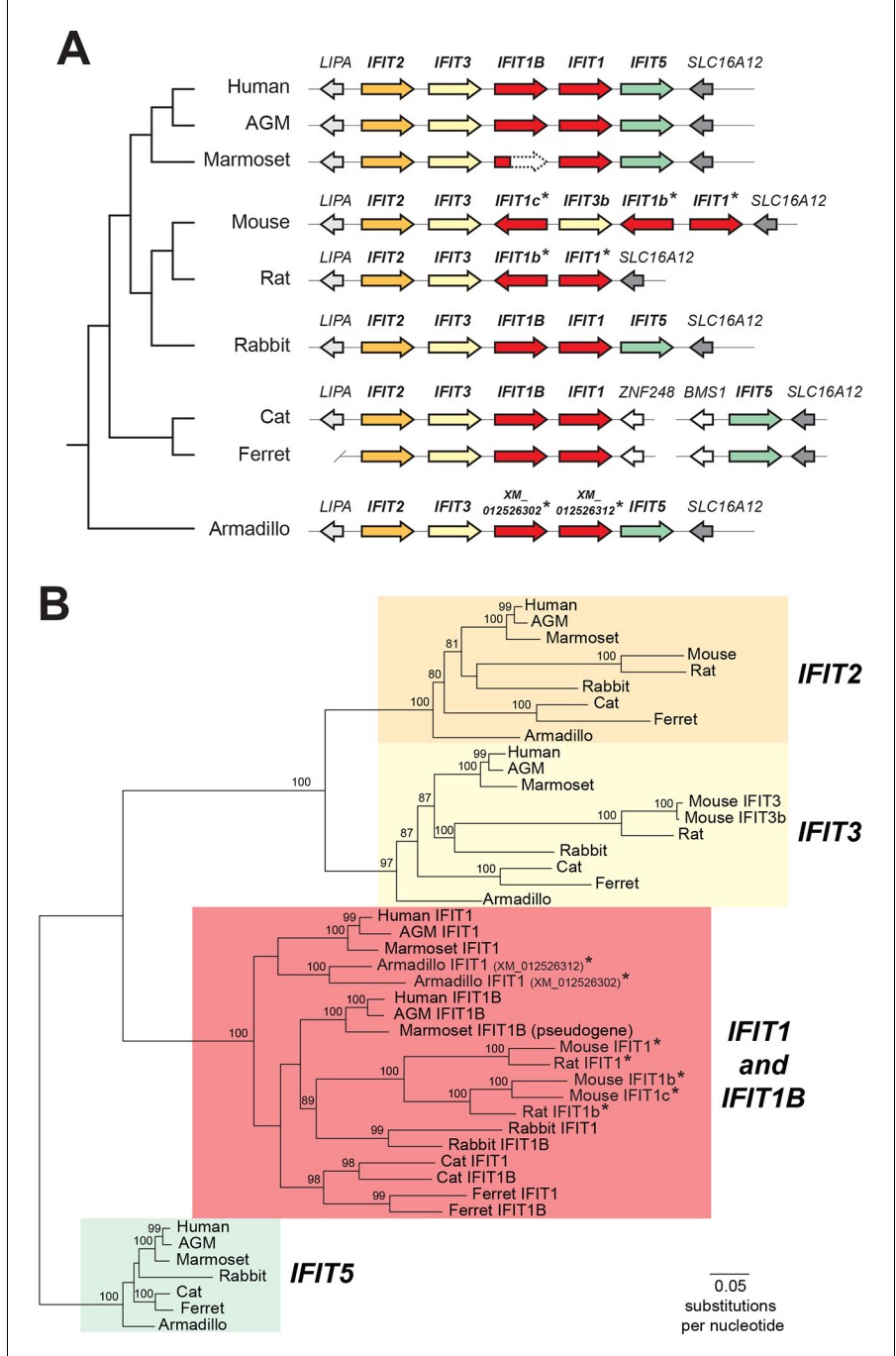

**Figure 1.** Discordant signatures of synteny and phylogeny for mammalian *IFIT* genes. (**A**) Alignment of the *IFIT* gene locus from several mammalian genomes. Colored arrows indicate intact *IFIT* genes and grey arrows indicate neighboring syntenic genes. At left is a schematic phylogenetic tree showing the relatedness of the indicated species. In carnivores such as cats and ferrets, the *IFIT* locus is split with *IFIT5* in one chromosomal location and the remainder of the *IFIT*s in another. In the marmoset genome, the *IFIT1B* gene has been pseudogenized by frameshift and nonsense mutations, as indicated by the dashed white arrow. Note that the nomenclature for the *IFIT* genes marked with asterisks is as previously proposed; we suggest a revised nomenclature scheme for those genes in this report from **Figure 2** onwards. A version of this figure, with revised gene names and coloring, is found in **Figure 2—figure supplement 4**. (**B**) Maximum likelihood phylogenetic tree generated using an alignment of the entire gene sequence of the indicated *IFIT*s. Bootstrap values greater than 80% are shown along the supported branch. Gene names with asterisks next to them have been revised from **Figure 2** onward. Accession numbers for all sequences are found in **Supplementary file 1**.

IFIT3, suggesting that a single duplication of IFIT1 may have occurred early in mammalian evolution, i.e., before the radiation of placental mammals over 100 million years ago. In primates, rabbits and carnivores, the paralogs immediately adjacent to IFIT3 are named IFIT1B in sequence databases, whereas the distal paralogs are named IFIT1. Based on this previous nomenclature, we will henceforth refer to these as the IFIT1 and IFIT1B gene families.

In contrast, the IFIT locus in mice and rats is arranged quite differently than other mammalian genomes. Instead of two paralogs, mice have three IFIT1 paralogs, two of which are in the opposite orientation to IFIT2 and IFIT3 (Figure 1A). Thus, shared synteny analyses are not suitable for assigning orthology/ paralogy relationships for the mouse IFIT1 genes. We, therefore, constructed maximum likelihood phylogenetic trees of full-length IFIT gene sequences across these divergent mammalian species to determine how mouse IFIT1 genes are related to those found in other species (Figure 1B). We found unambiguous phylogenetic signatures that delineate IFIT2, IFIT3 and IFIT5 genes into distinct, monophyletic clades that diverged as expected based on known mammalian species evolution (O'Leary et al., 2013). In contrast to these three gene families, we were unable to resolve IFIT1 and IFIT1B genes into distinct clades in our phylogenetic analyses. Instead, similar to previous studies (Liu, 2013), we found several instances of IFIT1 and IFIT1B from the same species (e.g., cat) appearing more phylogenetically related to each other than to their similarly named genes from a sister species (e.g., ferret) (Figure 1B).

The discordant phylogenetic signatures of IFIT1 and IFIT1B evolution could have two alternative explanations. The first is that species-specific duplication may have independently given rise to IFIT1 and IFIT1B genes in different genomes, as has been previously proposed (Liu et al., 2013). Such a duplication pattern might explain why cat IFIT1B is more phylogenetically related to cat IFIT1 than to IFIT1B genes from other genomes. However, such an explanation would require recurrent duplication of IFIT genes in many lineages into the same genetic location and orientation, and is thus unlikely.

We, therefore, considered an alternate explanation. In this alternative, IFIT1 and IFIT1B genes duplicated early in mammalian evolution but have recurrently recombined with each other. Resulting gene conversion would scramble the phylogenetic relatedness of the two genes without altering their genomic locations. Such gene conversion has been shown to contribute to the evolution of several multigene families including mammalian interferon alpha genes (Benovoy and Drouin, 2009; Hurles, 2004; Petronella and Drouin, 2011; Santoyo and Romero, 2005; Song et al., 2011; Woelk et al., 2007; Yasukochi and Satta, 2015). To address this possibility, we compared the sequences of IFIT1 and IFIT1B genes within species and between species to look for evidence of gene conversion. We found strong characteristic signatures of gene conversion frequently occurring between the 5' portions of IFIT1 and IFIT1B genes. For example, the sequences of IFIT1 and IFIT1B genes in cats are 86% identical at the whole gene level. However, these two genes are 97% identical in the 5' ends of the genes (nucleotides 1–948 in cat IFIT1) but only 63% identical in the 3' ends of the genes (Figure 2A). This dichotomy is apparent even if we restrict the analysis to just synonymous substitutions, where we observe a much higher rate of synonymous substitutions in the 3' region of the alignment (Figure 2—figure supplement 1). Such an uneven distribution of sequence differences strongly suggests that the 5' ends of IFIT1 and IFIT1B underwent gene conversion recently in the evolution of the cat IFIT locus, leading to sequence homogenization of the first two-thirds of the gene. In contrast, comparing orthologous genes between species (e.g. IFIT1B between cat and ferret) reveals an even distribution of differences across the gene sequences (Figure 2B) indicating that both the 5' and 3' regions have diverged to the expected degree between species. Moreover, this pattern of gene conversion is seen in pairwise alignments of many IFIT1 and IFIT1B genes. For instance, between ferret IFIT1 and IFIT1B, the 5' ends are 98% identical, but 3' ends are only 62% identical (Figure 2B). A similar dichotomy is evident in a comparison between rabbit IFIT1 and IFIT1B and between armadillo IFIT1 and IFIT1B (Figure 2—figure supplement 1). Interestingly, all of these gene conversion events have a similar recurrent breakpoint approximately two-thirds of the way through the gene sequence (Figure 2—figure supplement 1) that maps to a 'pivot' point between subdomains in the homologous IFIT5 structure (Abbas et al., 2013) (Figure 2—figure supplement 2). In all cases, the 5' ends of the IFIT1 genes are largely homogenized, whereas the 3' ends appear to be consistently diverging. Thus, our findings suggest that recurrent gene conversion has confounded the phylogenetic relationships of the 5' segment of IFIT1 and IFIT1B, which likely do not accurately represent the actual ancestry of these genes.

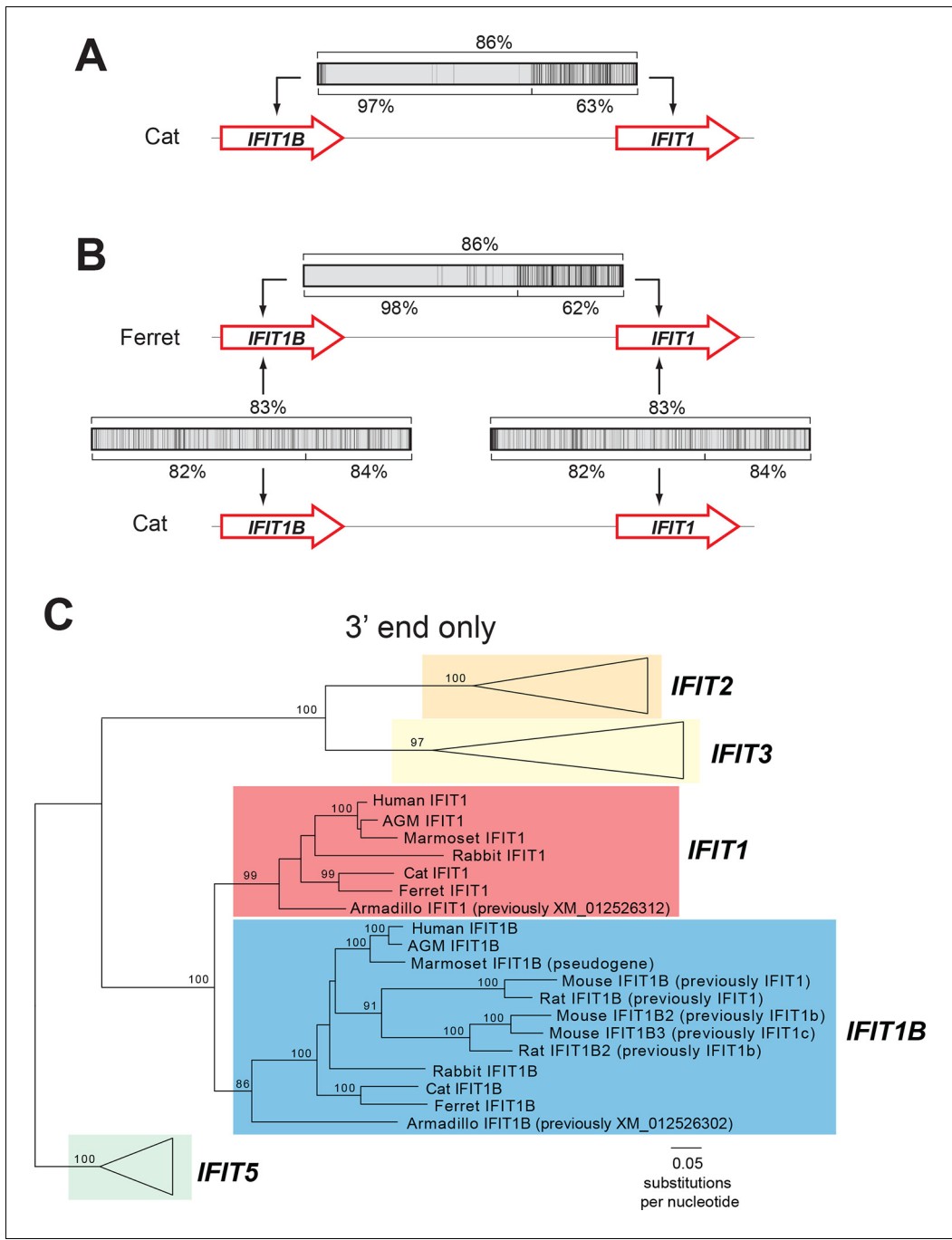

**Figure 2.** An ancient gene duplication gave rise to paralogous *IFIT1* and *IFIT1B* gene families. (**A**) Comparison of the nucleotide sequence of cat *IFIT1* and *IFIT1B* indicating gene conversion between the two paralogs. The middle rectangle represents a pairwise sequence alignment, with black vertical lines indicating differences between the two gene sequences. Percent nucleotide identity for the whole gene comparison is shown above the alignment schematic, with the 5' end and 3' end percent identities shown below. (**B**) Additional pairwise sequence alignments are shown as in part A. Additional examples supporting a common recombination breakpoint are shown in *Figure 2—figure supplement 1*. Mapping of the recombination breakpoint onto IFIT structural models (*Abbas et al., 2013*; *Yang et al., 2012*) is shown in *Figure 2—figure supplement 2*. (**C**) Maximum likelihood phylogenetic tree of *IFIT* genes using only the 3' end of the gene alignment (corresponding to bases 907–1437 of human *IFIT1*). Bootstrap values greater than 80% are shown along the supported branch. For clarity, *IFIT2*, *IFIT3* and *IFIT5* branches are collapsed and represented as triangles. A maximum likelihood phylogenetic tree of the 5'

*Figure 2 continued on next page*

*Figure 2 continued*

end of the *IFIT* gene alignment is shown in *Figure 2—figure supplement 3*. A figure showing revised gene names on the gene locus alignment is shown in *Figure 2—figure supplement 4*.

The following figure supplements are available for figure 2:

**Figure supplement 1.** Recurrent gene conversion across several diverse mammalian species has utilized a similar recombination breakpoint.

**Figure supplement 2.** Gene conversion of *IFIT1* and *IFIT1B* homogenizes some IFIT structural domains while leaving others to independently evolve.

**Figure supplement 3.** Phylogenetic tree of the recombining region of mammalian *IFIT* genes.

**Figure supplement 4.** Synteny of mammalian *IFITs* with updated *IFIT1* and *IFIT1B* gene names.

---

To determine the phylogenetic relationship of *IFIT1* and *IFIT1B* genes in the absence of gene conversion, we created a maximum likelihood phylogenetic tree of the same *IFIT* sequences shown in *Figure 1B*, but with just the nucleotide sequences 3' of the observed recombination breakpoint(s) (*Figure 2C*). In contrast to the analyses using the full-length *IFIT* genes (*Figure 1B*) or the 5' end of the *IFIT* genes (*Figure 2—figure supplement 3*), we now discern clear separation of *IFIT1* and *IFIT1B* genes. Both *IFIT1* and *IFIT1B* genes form distinct monophyletic clades that diverge according to mammalian species evolution, similar to other *IFIT* genes. Based on this phylogenetic concordance, as well as the agreement with synteny data, we infer that this phylogeny reflects the actual ancestry of *IFIT1* and *IFIT1B* genes in mammals. Importantly, these data indicate that *IFIT1* and *IFIT1B* duplicated early in mammalian evolution. Following this duplication, the 3' ends of the two genes have been diverging according to speciation events, whereas the 5' ends have recurrently recombined. These data also reveal that all three mouse *IFIT1* paralogs (previously named mouse *IFIT1*, *IFIT1b* and *IFIT1c*) and both rat paralogs (previously named rat *IFIT1* and *IFIT1b*) unambiguously group within the *IFIT1B* gene family. We, therefore, infer that mouse and rat have lost all copies of IFIT1, likely through a deletion of both IFIT1 and the neighboring IFIT5. Instead, mouse and rat genomes now bear recently duplicated copies of *IFIT1B*. As a result of our phylogenetic analyses, we henceforth refer to these genes as mouse/rat *IFIT1B* (previously mouse/rat *IFIT1)*, *IFIT1B2* (previously mouse/rat *IFIT1b*) and *IFIT1B3* (previously mouse *IFIT1c*) (*Figure 2C* and *Figure 2—figure supplement 4*). This recombination-aware phylogenetic approach thus reveals substantial changes in the *IFIT* gene repertoire even within this limited number of mammalian species.

## Duplication, loss and gene conversion repeatedly altered mammalian IFIT gene repertoires

To determine the broader impact of *IFIT* gene duplication, loss and conversion beyond our initial sampling of mammalian species, we assembled over 200 *IFIT* genes from 51 vertebrate species and conducted maximum likelihood phylogenetic analyses (*Figure 3—figure supplement 1*). We found that most placental mammals contain members of the *IFIT1, 1B, 2, 3 and 5* gene families (*Figure 3*). In contrast, marsupials encode a more limited set of *IFIT* genes including a single gene that predates the *IFIT2/IFIT3* duplication and a single gene that predates *IFIT1/IFIT1B* divergence (*Figure 3* and *Figure 3—figure supplement 1*). Consistent with other analyses (*Liu et al., 2013*; *Varela et al., 2014*), we also observed independent duplication of *IFIT* genes in fish, as well as in birds and monotremes (*Figure 3* and *Figure 3—figure supplement 1*). Based on these analyses, we infer that the human-like *IFIT* repertoire (*IFIT1, 1B, 2, 3 and 5*) was established when placental mammals diverged from non-placental mammals approximately 100–200 mya while other vertebrates have undergone lineage-specific expansions of *IFITs*.

Consistent with our smaller sample, we found that *IFIT1* and *IFIT1B* could not be resolved using the whole gene sequence, whereas *IFIT2, IFIT3 and IFIT5* could be easily separated into distinct monophyletic lineages. To formally test for recombination breakpoints between *IFIT1* and *IFIT1B*, we analyzed the complete set of full-length *IFIT* genes with GARD and SBP (*Kosakovsky Pond et al.,*

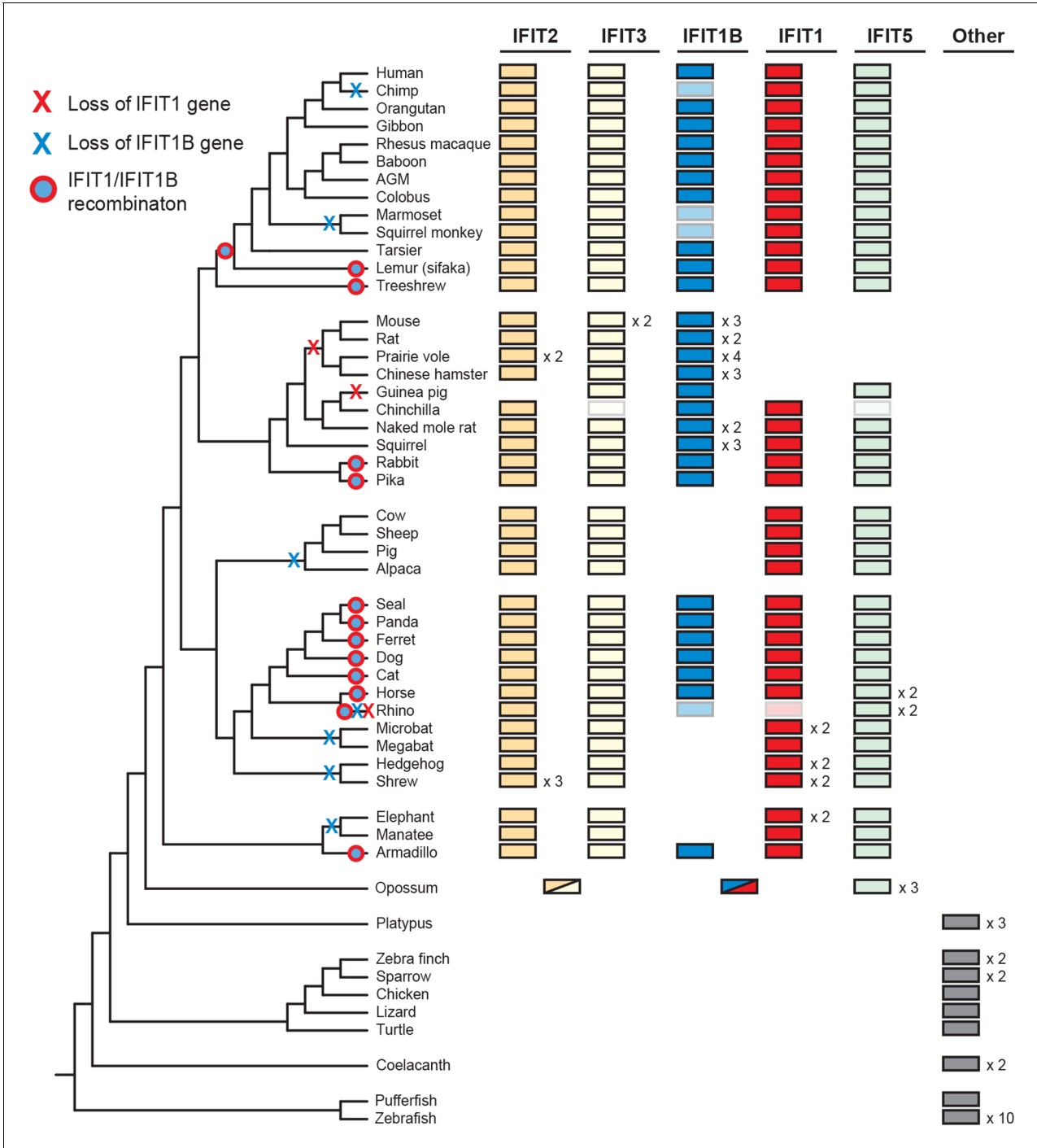

**Figure 3.** Widespread gene birth, gene loss and gene conversion in vertebrate *IFIT* genes. Summary table of the *IFIT* gene repertoire of 51 vertebrate species (see complete phylogeny in *Figure 3—figure supplement 1* as well as expanded regions of trees built from the 5' and 3' end of the *IFIT* gene alignment in *Figure 3—figure supplements 2* and *3*). Colored boxes indicate the presence of a given *IFIT* gene sequence, with multiple copies indicated as a number to the right of the colored box. Lightened boxes indicate that only a partial or pseudogenized copy of the gene can be found in the genome. On the phylogenetic tree to the left, X's indicate deletion or pseudogenization of either *IFIT1* or *IFIT1B* genes from that lineage (see *Figure 3—figure supplement 4* for primate examples). Also shown are predicted occurrences of gene conversion (red-around-blue circles) between *IFIT1* and *IFIT1B* sequences based on discordance between phylogenetic trees generated from 3' or 5' regions of the genes (for carnivore examples see *Figure 3—figure supplement 2*). Accession numbers for all sequences are found in *Supplementary file 1*.

The following figure supplements are available for figure 3:

*Figure 3 continued on next page*

*2006*), two tools that determine whether there is discordance between phylogenetic trees built from different regions of an alignment. Both analyses found strong statistical support (GARD p-value <0.001, SBP 100% model averaged support) for a recombination breakpoint approximately two-thirds of the way through the *IFIT* gene sequence (*Figure 3—figure supplement 2*). Importantly, performing the same analyses for the 3' region of the *IFIT* alignment found no evidence for additional recombination breakpoints (GARD p-value >0.1, SBP 0% model averaged support), supporting the assertion that these regions have not recombined during mammalian *IFIT* evolution. Indeed, construction of a phylogenetic tree of this non-recombining region of vertebrate *IFITs* shows that *IFIT1* and *IFIT1B* genes cluster into distinct monophyletic clades (*Figure 3—figure supplement 2* and *supplement 3*). In contrast, a phylogenetic tree created from just the 5' end of vertebrate *IFITs* reveals numerous instances of gene conversion, including at least five separate cases in the carnivores alone (*Figure 3—figure supplement 2*).

With this enhanced ability to assign *IFIT* genes to distinct *IFIT1* or *IFIT1B* families after separating the confounding effects of gene conversion, we were able to detect lineage-specific duplication and loss events. For instance, in primates, we found that *IFIT1* and *IFIT1B* genes have diverged as expected based on primate phylogeny and that all primates encode an intact *IFIT1*. In contrast, we found that *IFIT1B* has been pseudogenized at least twice in the primate lineage as the result of the introduction of nonsense codons and frameshift mutations, once in chimpanzees and again in both sequenced New World monkeys (*Figure 3—figure supplement 3* and supplement 4). Extending our surveys across all 51 sampled vertebrate species, we found numerous instances of *IFIT* gene birth, gene loss and gene recombination (*Figure 3*). In total, we found evidence for at least seven independent instances of *IFIT1B* loss, two separate instances of *IFIT1* loss, and at least 13 independent instances of *IFIT1/1B* gene conversion. Interestingly, it appears that most changes in the gene repertoire of *IFITs* across mammalian species are focused on *IFIT1* and *IFIT1B*, although there are also instances of lineage-specific gene births and loss in other IFIT genes as well. For example, *IFIT3* has duplicated in mice, whereas *IFIT5* has been lost in several rodents (which may have occurred coincident with *IFIT1* loss). These changes have led to a significant evolutionary turnover in the IFIT gene repertoire of different mammalian lineages, prompting us to investigate next the functional consequences of this evolutionary turnover of *IFIT1/ IFIT1B* paralogs in mammals.

## Discriminating IFIT1 and IFIT1B molecular functions using a novel budding yeast assay

The ancient duplication and retention of *IFIT1* and *IFIT1B* in most mammalian lineages suggested that they have non-redundant functions. Based on our phylogenetic findings, in addition to previous data suggesting human IFIT1 and mouse IFIT1B inhibit different viruses (*Andrejeva et al., 2013*; *Daffis et al., 2010*; *Pichlmair et al., 2011*; *Pinto et al., 2015*), we hypothesized that IFIT1 and IFIT1B might have evolved an alternative means of differentiating 'self versus non-self' and distinct antiviral specificities. We, therefore, investigated whether IFIT1 and IFIT1B proteins have distinct molecular discrimination properties.

Previous characterizations of mouse IFIT1B clearly established its role in recognizing the methylation status of 5' cap structure of mRNAs as a way to distinguish 'self', or host, mRNAs from 'non-self', or viral, mRNAs (*Daffis et al., 2010*; *Diamond, 2014*; *Fensterl and Sen, 2015*; *Hyde and Diamond, 2015*) (*Figure 4A*). However, most previous studies investigating mRNA cap requirements for IFIT1B-mediated growth inhibition have been carried out in the context of a virus infection under interferon-induced conditions. These approaches to understanding the function of individual IFIT

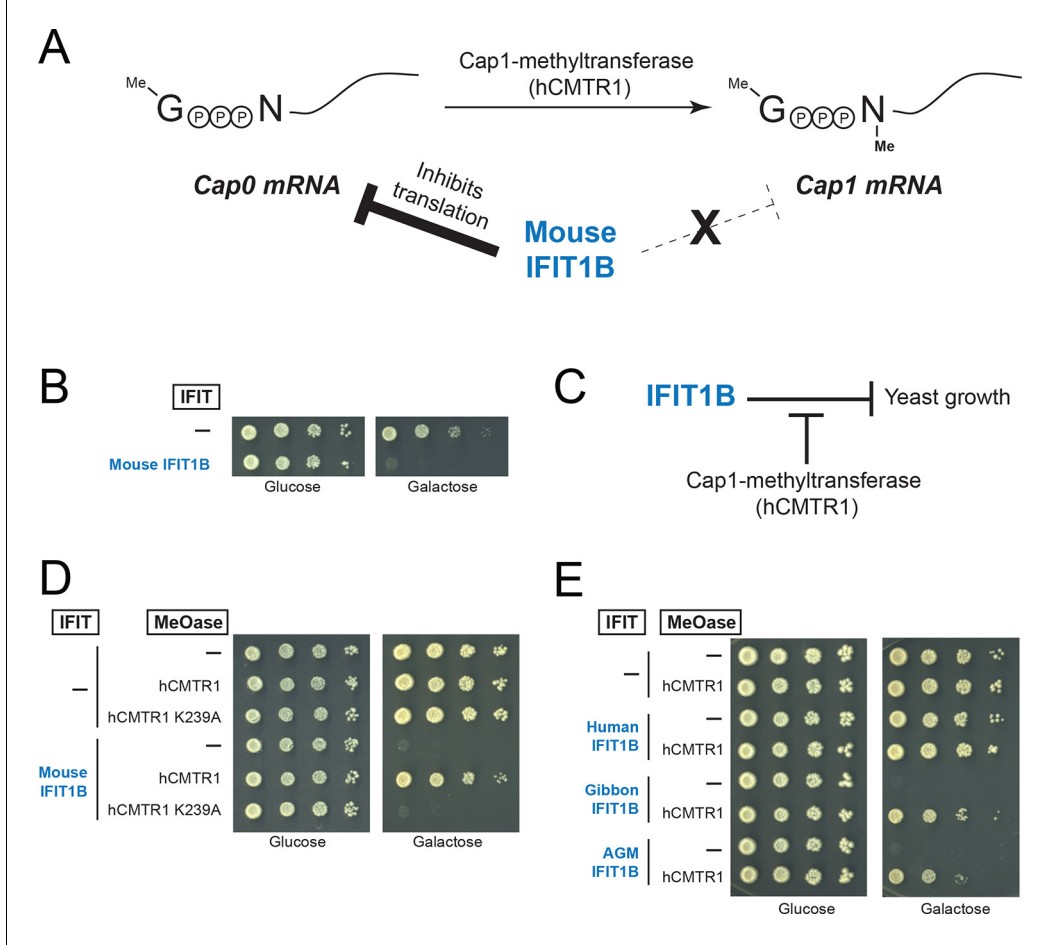

**Figure 4.** A yeast genetic assay recapitulates IFIT1B molecular specificity. (**A**) Schematic of mRNA cap structures and the proposed role of mouse IFIT1B in recognition and inhibition of mRNAs containing cap0-structures. Methylation of the first transcribed nucleotide of mRNAs, represented by an 'N', is performed by cap1-methyltransferases, such as human CMTR1 (hCMTR1). However, cap1-methyltransferase activity is absent in budding yeast. Mouse IFIT1B binds and inhibits translation of cap0-mRNAs, but its activity is blocked when mRNAs contain a cap1-structure. (**B**) A budding yeast growth assay for *IFIT1B* function. Since budding yeast lack a cap1-methyltransferase, their mRNAs are predicted to be targeted by mouse IFIT1B. Shown are 10-fold serial dilutions of yeast spotted on media in which the indicated *IFIT* is either not expressed (Glucose) or expressed (Galactose). Rows marked with '-' indicate an empty vector control. (**C**) Predicted role for hCMTR1 in blocking IFIT1B-mediated growth inhibition. (**D–E**) Yeast growth inhibition assays as in part B, but with galactose-induced co-expression of the indicated methyltransferase (cap1-methyltransferase (hCMTR1) wildtype or catalytic mutant (K239A)). In these panels, both IFIT and hCMTR1 expression are only induced upon grown on galactose-containing media. Protein expression data corresponding to these yeast strains grown in galactose are shown in *Figure 4— figure supplement 1*.

The following figure supplement is available for figure 4:

**Figure supplement 1.** Expression of IFIT1B and cap1-methyltransferase proteins in yeast.

proteins are potentially complicated by the induction of hundreds of other ISGs, as well as the fact several IFIT paralogs have been reported to bind to each other and function in a heterooligomeric complex (*Habjan et al., 2013*; *Pichlmair et al., 2011*).

We, therefore, wished to assay the specificity of IFIT1 and IFIT1B molecular discrimination in an orthogonal system. To this end, we turned to the budding yeast *Saccharomyces cerevisiae*, whose mRNAs only have cap0-structures (*Banerjee, 1980*) due to a lack of a cap1-methyltransferase. Based on this absence of a cap1-methyltransferase, we hypothesized that expression of mouse IFIT1B in

budding yeast might inhibit yeast growth just as IFIT1B inhibits replication of viruses with cap0-mRNAs. Indeed, using a galactose-induced IFIT1B expression system, we found that expression of mouse IFIT1B potently inhibits yeast growth (*Figure 4B*). Importantly, this growth inhibition occurs in the absence of any other IFIT or component of the innate immune system, indicating that mouse IFIT1B does not require any other mammalian-specific cofactor to perform this function. We next hypothesized that introducing the human cap1-methyltransferase hCRMT1 to yeast would rescue yeast growth by blocking the activity of IFIT1B (*Figure 4C*). Strikingly, we observed nearly complete rescue of yeast growth when we co-expressed the human cap1-methyltransferase (hCMTR1), but not

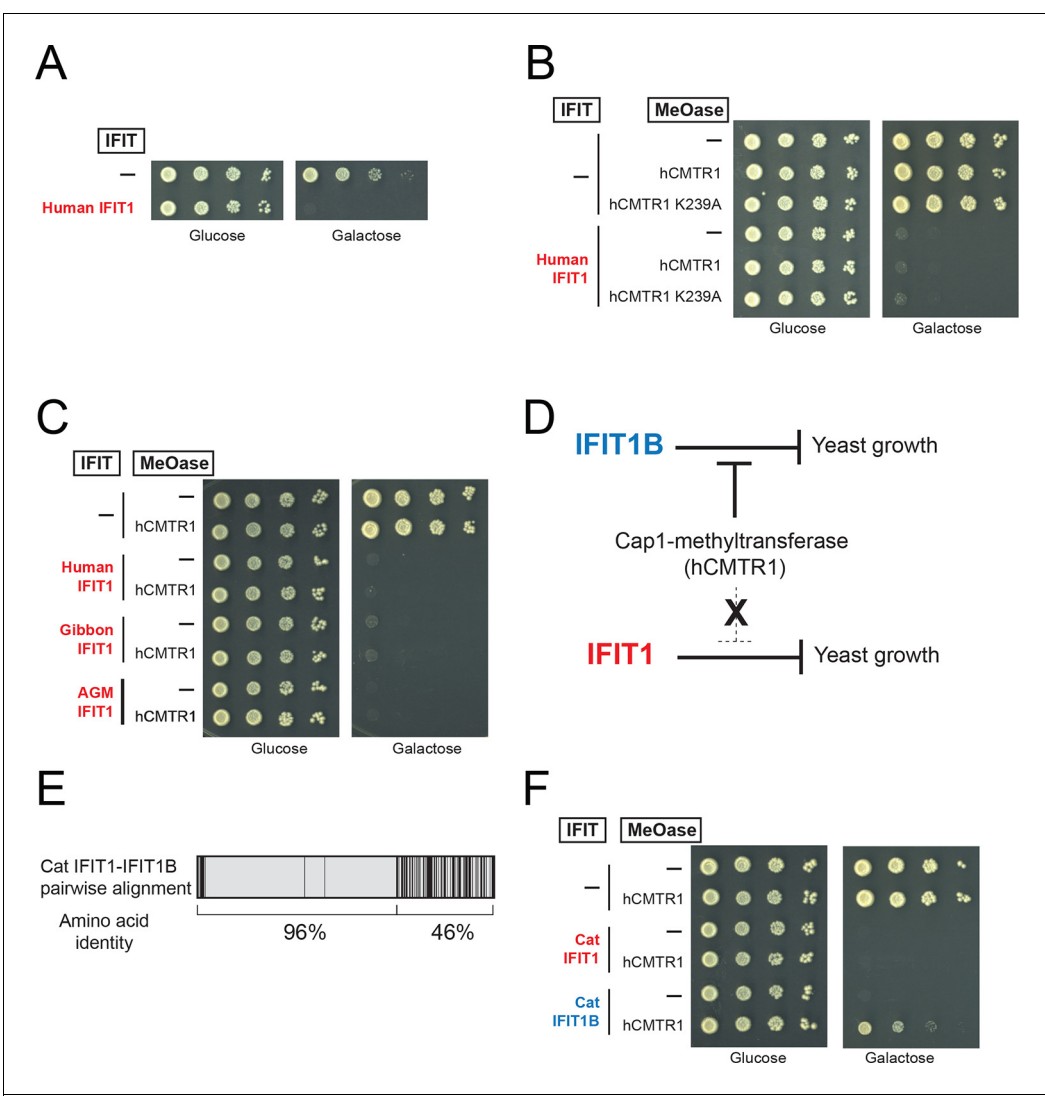

**Figure 5.** IFIT1-mediated growth inhibition is not relieved by a cap1-methyltransferase. (**A–C**) Yeast growth inhibition assays as in *Figure 4* with the indicated *IFIT1*. (**D**) In contrast to IFIT1B, IFIT1 inhibits yeast growth regardless of whether cap1-methyltransferase is present. (**E**) Pairwise comparison of the amino acid sequence of cat IFIT1 and IFIT1B, with black vertical lines indicating differences between the two protein sequences. Shown below are the pairwise amino acid identities for the N- and C-terminal regions of the alignment. (**F**) Yeast growth inhibition assays as in *Figure 4* with cat IFIT1 or IFIT1B. In these panels, both IFIT and hCMTR1 expression are only induced upon grown on galactose-containing media. Protein expression data corresponding to these yeast strains grown in galactose are shown in *Figure 5—figure supplement 1*.

The following figure supplement is available for figure 5:

**Figure supplement 1.** Expression of IFIT1 proteins in yeast.

a catalytic mutant of hCMTR1 (K239A) (*Belanger et al., 2010*) (*Figure 4D*). We, therefore, conclude that the mouse IFIT1B-mediated growth inhibition of budding yeast is specifically due to the lack of a functional cap1-methyltransferase in yeast, accurately recapitulating the known antiviral molecular specificity of IFIT1B (*Figure 4C*).

Based on our phylogenetic predictions, we next investigated whether other members of the *IFIT1B* gene family have similar properties as the well-characterized mouse IFIT1B. If so, we predicted that they would also cause growth inhibition of yeast, but be rescued by overexpression of the human cap1-methyltransferase. To test this hypothesis, we expressed IFIT1B from several primate species with or without co-expression of hCMTR1 (*Figure 4E*). Consistent with our phylogenetic prediction, we observed robust growth inhibition as a result of expression of gibbon and African green monkey (AGM) IFIT1B that could be rescued by expression of hCMTR1. Thus, despite the long divergence separating mouse and primate *IFIT1B* genes, they share molecular discrimination properties. Surprisingly, we observed no growth inhibition by human IFIT1B. This lack of activity even upon equivalent expression in yeast (*Figure 4—figure supplement 1*) is unusual among intact primate IFIT1Bs, suggesting that human IFIT1B lacks function. This result is consistent with previous suggestions (*Fensterl and Sen, 2011*) that human IFIT1B may be non-functional even though it appears to be encoded by an intact open reading frame.

We next tested our hypothesis that IFIT1 has evolved a distinct molecular function from that of IFIT1B. Similar to mouse IFIT1B, we found that expression of human IFIT1 resulted in a potent inhibition of yeast growth (*Figure 5A*). However, in contrast to mouse IFIT1B, we observed no rescue of human IFIT1-mediated growth inhibition upon overexpression of hCMTR1 (*Figure 5B*). IFIT1 proteins from gibbon and African Green Monkey (AGM) recapitulate the findings from human IFIT1 (*Figure 5C*). These data indicate that IFIT1 proteins do not distinguish between absence or presence of a cap1-methyltransferase (*Figure 5D*) but instead, likely recognize another as-yet-undefined 'non-self' molecular pattern, other than Cap0-mRNA, which is present in yeast.

These yeast genetic data indicate that human IFIT1 and mouse IFIT1B have evolved distinct molecular functions. As described above, resolution of *IFIT1* and *IFIT1B* genes into distinct monophyletic clades required considering only the non-recombining 3′ end of the genes (*Figure 2*). We, therefore, asked whether differences in the C-terminal end of the IFIT1 and IFIT1B were responsible for changes in their molecular specificity, as suggested by our phylogenetic analyses. Our initial chimeras between mouse IFIT1B and human IFIT1, or gibbon IFIT1 and IFIT1B, were unable to recapitulate the yeast growth inhibition. However, this result may be expected, as each of these proteins has diverged without gene conversion for >30 million years and may have acquired epistatic interactions between the N-terminus and the C-terminus.

As an alternate strategy to test whether the C-terminus is responsible for differences between IFIT1 and IFIT1B function, we turned to a natural pair of IFIT1/IFIT1B proteins in which the N-termini had recently undergone gene conversion. Cat IFIT1 and IFIT1B are 96% identical at the amino acid level in the N-terminus of the protein whereas the C-terminus is only 46% identical (*Figure 5E*). Upon overexpression in yeast, both cat IFIT1 and IFIT1B were capable of causing growth inhibition (*Figure 5F*). However, only inhibition by cat IFIT1B was rescued by hCMTR1, suggesting that the C-terminus of the IFIT1 paralogs is critical for the differences in the molecular specificity of these two proteins. We, therefore, conclude that *IFIT1* and *IFIT1B* gene families perform distinct molecular functions in the species in which they are found and that recurrent gene conversion of the N-termini of *IFIT1/IFIT1B* genes in mammals has not led to a loss of their separate functions.

## IFIT1 and IFIT1B encode distinct antiviral specificities

We showed that IFIT1 and IFIT1B proteins are phylogenetically distinct and are functionally distinct in their dependence on cap1-methyltransferase in our yeast assay. Next, we hypothesized that they have non-overlapping antiviral activities. To test this hypothesis, we created cell lines that constitutively overexpress different *IFIT1* or *IFIT1B* genes (*Figure 6A*), circumventing the need to stimulate *IFIT* expression with interferon and allowing us to evaluate different *IFIT1* or *IFIT1B* gene functions in an isogenic background. We then tested whether either human IFIT1 or mouse IFIT1B can specifically inhibit replication of a virus lacking a functional cap1-methyltransferase. For these experiments, we compared replication of a wildtype (cap1-mRNA expressing) vaccinia virus to replication of a vaccinia virus that has a catalytic mutation in its cap1-methyltransferase (*Latner et al., 2002*). This assay allowed us to test whether cap1-methyltransferase could enable a virus to block IFIT-mediated

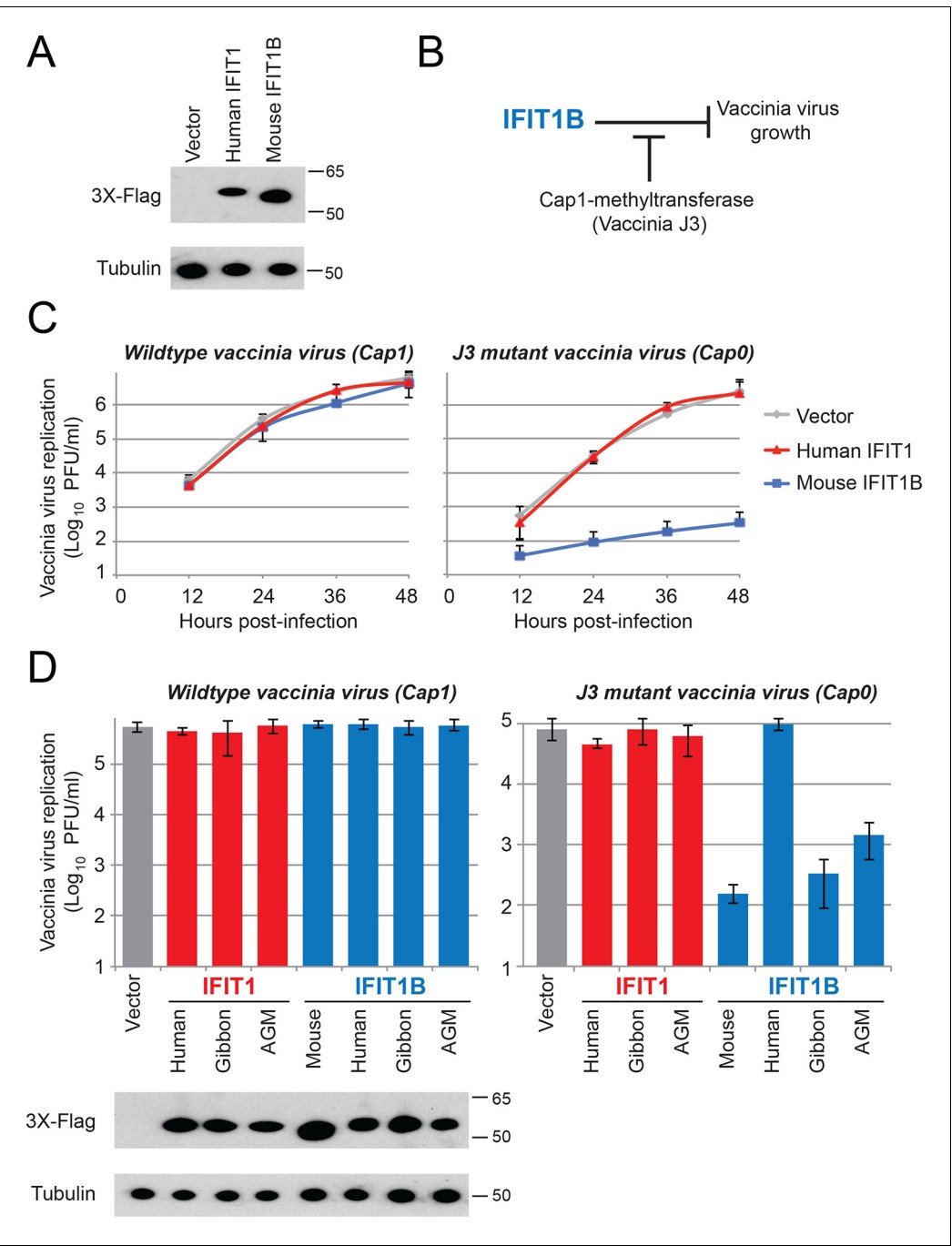

**Figure 6.** IFIT1B, but not IFIT1, inhibits replication of a cap0-containing virus. (**A**) Cell lines were generated that express human IFIT1 or mouse IFIT1B, as assessed by western blot. (**B**) Predicted role for vaccinia virus cap1-methyltransferase (J3) in blocking IFIT1B-mediated viral inhibition. (**C**) Cell lines expressing the indicated IFIT protein were infected with either wildtype (left) or cap1-methlytransferase mutant (J3 K175A, right) vaccinia virus at an MOI of 0.01. Virus was harvested at the indicated time point and titered. Error bars represent standard deviation of three biological replicates. (**D**) As in panel C, but only showing data 24 hr post-infection with wildtype (left) or the cap1-methyltransferase mutant (right) vaccinia virus. Shown below are protein expression levels from uninfected cell lines.

antiviral activity (*Figure 6B*). Consistent with previous studies with these same viruses (*Daffis et al., 2010*), we found that expression of mouse IFIT1B can potently inhibit replication of a mutant vaccinia virus that lacks cap1-methyltransferase activity, but was unable to inhibit replication of wildtype vaccinia virus (*Figure 6C*). While the previous mouse knockout experiment had demonstrated that mouse IFIT1B is necessary for inhibition of viruses encoding cap0-mRNAs (*Daffis et al., 2010*), our findings show that singular overexpression of mouse IFIT1B is also sufficient for such inhibition. We next extended our analyses to other members of the *IFIT1* and *IFIT1B* gene families to determine if cap0-mRNA dependence was a conserved feature of their antiviral activity. Consistent with our yeast assay data, we found that gibbon and AGM IFIT1B proteins function similarly to mouse IFIT1B, and can specifically inhibit the replication of a virus expressing cap0-mRNAs (*Figure 6D*). Also consistent with our analyses of IFIT1B in yeast (*Figure 4E*), human IFIT1B lacks antiviral activity despite being expressed in these cell lines. These results suggest that most IFIT1B proteins (except for the putatively non-functional human IFIT1B) share cap0-mRNA recognition as a discriminant of their antiviral function.

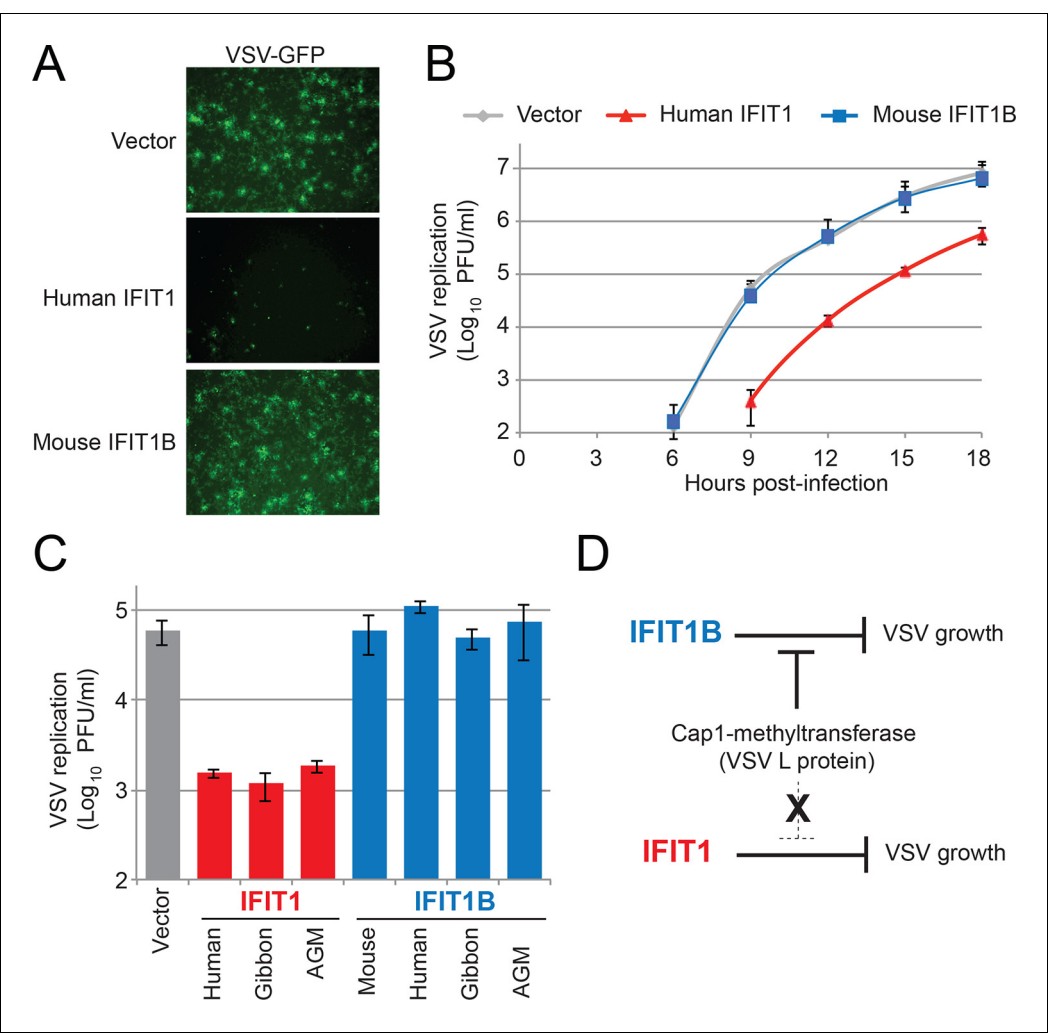

**Figure 7.** IFIT1, but not IFIT1B, inhibits replication of VSV, a cap1-containing virus. (A) Cell lines were infected with vesicular stomatitis virus (VSV) encoding a GFP protein at an MOI of 0.01. After 12 hr, cells were visualized for expression of the virally-encoded GFP. (B) Infections were performed as in part A, but at the indicated timepoint, virus was harvested and titered. Error bars represent standard deviation of three biological replicates. (C) Infections were performed as in part A, harvested after 12 hr, and titered. Error bars represent standard deviation of three biological replicates. (D) In contrast to IFIT1B, IFIT1 inhibits VSV growth arrest regardless of the fact that cap1-methyltransferase is present.

In contrast to the activity of most IFIT1B proteins, human IFIT1 did not inhibit replication of the vaccinia virus methyltransferase mutant (*Figure 6C*). Consistent with our yeast data, these results demonstrated that human IFIT1 does not discriminate 'self versus non-self' based on cap0-mRNA versus cap1-mRNA specificity. Likewise, neither gibbon nor AGM IFIT1 was able to inhibit the cap0-mRNA expressing vaccinia virus (*Figure 6D*). We, therefore, conclude that in keeping with their phylogenetic divergence, IFIT1B and IFIT1 lineages evolved to rely on distinct cues for their function. Such non-redundancy could explain the retention of both IFIT1 and IFIT1B in most mammalian genomes.

These results also suggest that there likely exist viruses that are specifically inhibited by IFIT1 but not IFIT1B. To identify such viruses, we took advantage of previous studies, which showed that knockdown of human IFIT1 can augment replication of several viruses expressing a functional cap1-methyltransferase (*Andrejeva et al., 2013*; *Pichlmair et al., 2011*). Based on our findings and others (*Daffis et al., 2010*; *Hyde et al., 2014*; *Ma et al., 2014*; *Menachery et al., 2014*; *Szretter et al., 2012*; *Zust et al., 2011*; *Habjan et al., 2013*; *Kimura et al., 2013*; *Kumar et al., 2014*) we expect that all cap1-mRNA expressing viruses would be resistant to IFIT1B restriction. We, therefore, infected our cell lines expressing either human IFIT1 or mouse IFIT1B with vesicular stomatitis virus (VSV), a virus known to express mRNAs with a cap1-structure (*Ma et al., 2014*; *Banerjee, 1980*). As expected, we found that mouse IFIT1B had no effect on either virally-encoded GFP expression (*Figure 7A*) or viral titers (*Figure 7B*). VSV viral titers were also unaffected by other IFIT1B genes (*Figure 7C*). In contrast, we observed a substantial decrease in both viral-GFP expression and viral titer in cells expressing human IFIT1 (*Figure 7A,B*). This inhibitory activity was also conserved in other primate IFIT1 proteins (*Figure 7C*), suggesting that multiple IFIT1 proteins are insensitive to cap1-methyltransferase and share the property of inhibiting VSV (*Figure 7D*). Together with our vaccinia virus data (*Figure 6*), these data support the conclusion that IFIT1B and IFIT1 encode non-redundant antiviral specificities that together expand the antiviral range of the IFIT gene repertoire in mammals.

## Discussion

Rapid gene evolution is a hallmark of antiviral proteins. Immunity proteins are driven to innovate continually to maintain effective defenses against an evolving barrage of pathogens (*Daugherty and Malik, 2012*). One common genetic mechanism of innovation employed by host genomes is gene duplication, which enables hosts to either escape viral antagonism or evolve new inhibitory mechanisms through subfunctionalization. Indeed, many antiviral genes expressed as part of the interferon response represent members of multigene families, whose lineage-specific expansions and contractions have shaped contemporary repertoires of antivirals in mammalian genomes (*Brunette et al., 2012*; *Daugherty and Malik, 2012*; *Munk et al., 2012*; *Tareen et al., 2009*; *Zhang et al., 2012*). In several of these multigene families, gene conversion has further diversified the innate immune repertoire of different species (*Buchmann, 2014*; *Mitchell et al., 2015*; *Woelk et al., 2007*).

Despite being among the most highly expressed interferon-stimulated genes (ISGs), the function of the IFIT antiviral genes is still incompletely understood. Previously, several IFITs, including mouse IFIT1B (previously IFIT1) and human IFIT1, were shown to inhibit protein synthesis (*Guo et al., 2000*; *Hui et al., 2003*; *Terenzi et al., 2005*). These findings are consistent with mRNA translation control being a major axis of regulation of viral replication (*Mohr and Sonenberg, 2012*; *Li et al., 2015*). Furthermore, the finding that mouse IFIT1B represses the translation of mRNAs lacking Cap1-structures appeared to resolve the conundrum of how at least this IFIT protein discriminates between 'self' and 'non-self' (*Daffis et al., 2010*). However, the broad application of this result for IFIT1 function was complicated by apparently contradictory findings that mouse and human representatives of IFIT1 have different antiviral specificities (*Andrejeva et al., 2013*; *Daffis et al., 2010*; *Pichlmair et al., 2011*; *Pinto et al., 2015*). Thus, it remained unclear how different IFIT proteins recognize and inhibit replication of different viruses, and whether and how the changes in IFIT gene composition between species (e.g. humans and mice) have altered their antiviral repertoire.

To provide an evolutionary framework for understanding how the IFIT antiviral repertoire evolved in mammals, we performed in-depth phylogenetic analyses. We find unambiguous evidence that, although they were initially assumed to be orthologs, mouse IFIT1B (previously mouse *IFIT1*) and human *IFIT1* represent paralogous genes that diverged close to the origin of placental mammals.

We further showed that the *IFIT* gene family, especially *IFIT1* and *IFIT1B* genes, have undergone recurrent bouts of gene duplication, gene loss, and gene conversion. These processes have resulted in a wide diversity of *IFIT* genes across mammalian species. Such changes in *IFIT* gene composition might be expected to affect the range of viruses that can be inhibited by IFITs in different mammalian genomes.

Based on their long divergence, we predicted that mouse (and other) IFT1B and human (and other) IFIT1 protein might possess different antiviral specificities. Consistent with this prediction, our data indicate that only IFIT1B proteins distinguish 'self from non-self' mRNAs by recognizing an unmethylated (cap0-) mRNA structure, a molecular pattern that is absent on 2'O-methylated (cap1-) host mRNAs. However, the general antiviral effectiveness of IFIT1B proteins is limited due to the fact that every mammalian virus family, except alphaviruses, has evolved a way to produce cap1-mRNAs. Viral cap1-mRNAs are produced either by nuclear transcription and use of the host nuclear capping machinery (e.g., retroviruses and many DNA viruses), or cap-snatching (e.g., orthomyxoviruses), or virally encoding cap1-methyltransferases (e.g., poxviruses and rhabdoviruses) (*Banerjee, 1980*; *Decroly et al., 2012*; *Hyde and Diamond, 2015*). Even in the case of alphaviruses, which produce cap0-mRNAs (*Banerjee, 1980*) and are therefore predicted to be susceptible to IFIT1B-mediated inhibition, secondary structure changes at the 5' end of alphaviral mRNAs can blunt IFIT1B antiviral action (*Hyde et al., 2014*). Given all of these viral counter-strategies, it might appear that IFIT1B is on the losing side of the arms race between host and most viruses. These counter-strategies may partially explain why we find that *IFIT1B* has been deleted or pseudogenized at least seven separate times in mammalian evolution. Even in humans, which encode a full-length *IFIT1B* ORF, IFIT1B activity appears to have been lost both due to lack of interferon-inducibility (*Fensterl and Sen, 2011*) as well as loss of protein function (*Figure 4E* and *6D*).

Humans and most other mammalian species (excluding rodents) do, however, have an intact member of the *IFIT1* gene family. IFIT1 mechanism and antiviral specificity have been enigmatic in part due to incorrect comparisons with mouse IFIT1B. By experimentally separating the functions of IFIT1 and IFIT1B from the rest of the innate immune system, we now show that IFIT1 has evolved a molecular specificity that differs from IFIT1B and is not blocked by cap1-methyltransferase. For example, using a yeast genetic assay, we show that IFIT1, unlike IFIT1B, potently inhibits yeast growth independent of whether human cap1-methyltransferase is co-expressed. Moreover, IFIT1, but not IFIT1B, inhibits replication of VSV, a virus encoding a cap1-methyltransferase. These results establish that these two paralogous immunity factors have distinct molecular function and highlight a significant role for IFIT1 in specific antiviral defense that is distinct from IFIT1B.

Our study does not elucidate the molecular means by which human IFIT1 inhibits viral replication. While several possibilities exist for how IFIT1-mediated translational repression might be controlled, we favor the possibility that IFIT1, like its paralog IFIT1B, distinguishes a 'self versus non-self' pattern on mRNA to selectively inhibit viral replication. Supporting this model is the fact that in addition to IFIT1B, IFIT2 and IFIT5 have been reported to have sequence- or structure-specific binding to RNA (*Habjan et al., 2013*; *Katibah et al., 2013*; *2014*; *Yang et al., 2012*), suggesting that numerous IFITs may restrict viral replication through recognition of distinct viral RNA patterns. Although the presumed molecular pattern that IFIT1 recognizes remains unknown, our data suggest that mammalian hosts, as well as vaccinia virus, possess a 'self' molecular pattern to prevent IFIT1-mediated inhibition whereas yeast and VSV display a 'non-self' molecular pattern that cannot block IFIT1 action. One previous proposal suggested that human IFIT1 antiviral activity resulted from direct binding and sequestration of 5' triphosphate ends of viral replication intermediates, thus interfering with replication rather than directly inhibiting translation (*Pichlmair et al., 2011*). We instead favor a model in which IFIT1 recognizes a distinct 'self versus non-self' pattern to inhibit mRNA translation directly, similar to IFIT1B. Biochemical data suggests that IFIT1, like IFIT1B, binds to cap-proximal RNA and prevents binding of mRNAs by the translation initiation factor eIF4F (*Kumar et al., 2014*). Moreover, human IFIT1 was shown to not just inhibit replication of PIV5, but to inhibit translation of PIV5 2'O-methylated mRNAs (*Andrejeva et al., 2013*). Finally, our observation that IFIT1 expression also inhibits yeast growth suggests that it is not a viral replication intermediate, but rather a 'non-self' pattern, which is recognized by IFIT1, similar to the lack of mRNA 2'O-methylation for IFIT1B. While the methylation state of the first transcribed nucleotide (cap0-structure versus cap1-structure) is not the determinant of IFIT1 specificity, other chemical modifications near the cap, or mRNA sequence

determinants, may allow IFIT1 to distinguish 'self' from 'non-self' to selectivity inhibit viral replication.

Although gene conversion and gene turnover initially confounded *IFIT* phylogenetic analysis, it may now present a significant opportunity to understand the biochemical basis of the different antiviral specificities of IFIT1 and IFIT1B. For instance, two cat IFIT proteins, which are practically identical in their N-terminal two-thirds, but divergent in their C-terminus, fully recapitulate the expected properties of IFIT1 and IFIT1B in our yeast assay. Use of other such 'natural' chimeras can help guide the biochemical dissection of what provides IFIT1B with its unique cap0-mRNA recognition properties and provide insight into how IFIT1 has evolved its distinct specificity. The naturally occurring recombination breakpoint in the IFIT1/1B gene conversion tracts also suggests features of IFITs that are critical for a common aspect of IFIT1/1B activity versus those that promote their distinct molecular specificity. In the structures of IFIT5 bound to the 5' triphosphate containing end of RNA, three 'subdomains' surround the RNA (*Abbas et al., 2013*). The recombination breakpoint we observed falls within the 'pivot' region separating the second and third subdomain (*Figure 2—figure supplement 2*), suggesting the first two subdomains may be important for a common conserved function of IFIT1 and IFIT1B whereas the third subdomain might confer changes to the molecular specificity. We also note that the breakpoint occurs just outside of the homooligomerization domain in the structure of IFIT2 (*Yang et al., 2012*). As some IFITs are thought to oligomerize for function (*Habjan et al., 2013*; *Pichlmair et al., 2011*), the homogenization of the region corresponding to the oligomerization domain in IFIT2 suggests that IFIT1 and IFIT1B might also need to hetero-oligomerize in species in which both are present.

Based on our new understanding of the divergent functions of IFIT1 and IFIT1B, we hypothesize that duplication or loss of different *IFIT* genes might directly influence the ability of a particular host species to defend against specific viral pathogens. For instance, the duplication of *IFIT1B* genes in rodents might have been driven by pressure from alphaviruses or other undiscovered cap0-mRNA expressing viruses, and may augment the defenses of rodent hosts against such viruses. In contrast, the loss of *IFIT1* in many rodents might leave them more susceptible to infection by viruses inhibited by IFIT1, such as VSV. Similarly, loss of IFIT1B function in a broad range of species, including humans, may result in greater ability of cap0-mRNA expressing viruses to replicate in those species. In that regard, rodents may be better prepared than humans, chimpanzees, New World monkeys and several other species to defend against cap0-mRNA expressing viruses such as alphaviruses (*Figure 3*). Thus, *IFIT* gene duplication may be the host's response to the evolutionary arms race where viruses are continually masking their RNAs with 'self' features. On the other hand, when the selective pressure from viruses displaying a given 'non-self' feature (e.g. Cap0-mRNAs) has been relaxed, the specific *IFIT* gene that restricts those viruses may be pseudogenized or lost permanently. Although compensatory mechanisms may have evolved that mitigate the consequences of loss of a single IFIT, our results indicate that lineage-specific loss of *IFIT* genes eliminates an elegant means to discriminate 'self versus non-self' RNAs from the host's armament. We predict that the dynamic evolution of IFIT genes across mammals will have important consequences for species-specific antiviral immunity.

## Materials and methods

### Phylogenetic analyses

Mammalian *IFIT* genes were identified in genomes of the indicated species using human IFIT protein sequences to query the non-redundant (nr) database using tBLASTn (*Altschul et al., 1997*) (see *Supplementary file 1* for accession numbers). In human *IFIT* genes, the entire protein-encoding sequence except for the N-terminal methionine is encoded on a single exon. For the species indicated in *Figure 1*, we used this information to include only *IFIT* sequences that were found on a single uninterrupted region of DNA in either NCBI or UCSC genome databases. For the broader panel of species shown in *Figures 2* and *3*, several of the genomes are not as well assembled and we therefore did not eliminate genes based on these criteria but insisted that >90% of the gene sequence be available in NCBI. *IFIT* sequences were aligned based on their translated sequence using MUSCLE (*Edgar, 2004*) implemented in Geneious (*Kearse et al., 2012*). All alignments were manually curated using Geneious (*Kearse et al., 2012*). Maximum likelihood phylogenetic trees of *IFIT* nucleotide sequences were generated using the HKY85 substitution model in PhyML

(*Guindon et al., 2010*) using 1000 bootstrap replicates for statistical support. K-estimator (*Comeron, 1999*) was to calculate the rate of synonymous change (dS) for each 200 nt window of pairwise comparisons of *IFIT1* and *IFIT1B* genes. To examine the alignments for evidence of recombination breakpoints, we used the SBP and GARD algorithms implemented at DataMonkey.org (*Kosakovsky Pond et al., 2006*). Pairwise percent identity calculations and graphical representations were made using Geneious (*Kearse et al., 2012*). Phylogenetic trees were visualized using FigTree (http://tree.bio.ed.ac.uk/software/figtree/). Protein structural figures were generated using PyMol (https://www.pymol.org).

## Construction of yeast strains

Previous studies have utilized budding yeast as a genetic assay system for PKR-mediated mRNA translation inhibition by showing that *PKR* expression in yeast arrests growth (*Dever et al., 1993*). We therefore asked whether expression of *IFIT* genes in yeast would also arrest growth. *IFIT* genes with an N-terminal 3xFlag tag were cloned downstream of the Gal1-10 promoter in the Cen-based plasmid, p413, using primers described in *Supplementary file 2*. Yeast (strain BY4741) were transformed and selected on synthetic complete media lacking histidine and containing 2% glucose (SC -his GLU). For expression of methyltransferases and IFITs, human methyltransferases were integrated into strain BY4741 to replace the His3 gene and then transformed with p413 plasmids expressing IFITs. First, the human genes for hCMTR1 or hCMTR1 K239A methyltransferases were cloned downstream of the Gal1-10 promoter of pRS305-Gal1 using primers described in *Supplementary file 2*. The resulting plasmids were used as templates to amplify the entire region spanning the Gal promoter to the Leu2 gene using primers described in *Supplementary file 2* with 70 bp homology to the genomic regions flanking the His3 gene. The resulting PCR products were transformed into strain BY4741 and yeast were selected on media lacking leucine (SC -leu GLU). Yeast with integrated galactose-inducible methyltransferases were subsequently transformed with IFIT genes as described above and selected on media lacking both leucine and histidine (SC -leu/-his GLU).

## Yeast growth assays

All yeast plating assays were performed on selective media with either 2% glucose (for uninduced controls) or 2% galactose (for induction of IFIT and methyltransferase genes). A single yeast colony was picked from a freshly streaked plate of transformed yeast and grown overnight at 30 degrees in liquid media (SC -his GLU or SC -his/-leu GLU). Saturated overnight cultures were serially diluted 10-fold from a starting $OD_{600}$ of 1 and spotted onto selective plates containing glucose or galactose. Plates were incubated at 30 degrees for 48–72 hr.

## Construction of cell lines expressing IFITs

The retroviral packaging vector pQCXIP (Clontech, Mountain View, CA) was used as a backbone for generation of all stable cell lines. First, an insert containing mCherry followed by a T2A site followed by a 3xFlag tag was cloned downstream of the CMV promoter of pQCXIP, resulting in plasmid pMD143 (see *Supplementary file 2*). All *IFIT* genes were inserted in frame with the 3xFlag tag of pMD143 using primers and templates described in *Supplementary file 2*. The resulting plasmids were used to generate VSV-g pseudotyped retroviruses that were then used to transduce BSC40 cells, an African green monkey derived kidney epithelial cell line. The BSC40 cells have been routinely screened for mycoplasma infections and validated as being of African Green monkey origin using RT-PCR analyses. We also note that BSC40 cells are not on the list of commonly misidentified cell lines maintained by the International Cell Line Authentication Committee. Stably transduced BSC40 cells were selected and maintained in culture media (DMEM with 10% FBS) containing 10 ug/ml puromycin. All cell lines were assayed within the first 20 passages following transduction.

## Viral infectivity assays

Vaccinia virus strain WR wildtype or cap1-methyltransferase mutant (J3 K175R) were a generous gift of R. Condit (*Latner et al., 2002*). VSV-GFP was a generous gift of J. Rose (*Boritz et al., 1999*). For infectivity assays, BSC40 cells stably transduced with pMD143 containing various *IFIT* gene inserts were seeded to 24-well plates (~50000 cells/well) and allowed to grow overnight. Cells were incubated with vaccinia virus or VSV-GFP at a multiplicity of infection (MOI) of 0.01 for two hours,

followed by a media change. Vaccinia virus was harvested at the indicated time post-infection by freeze/thaw disruption of cells. VSV-GFP was harvested from cell supernatant at the indicated time post-infection. Viruses were titered on BSC40 cells either with agar overlays (for VSV-GFP) or without (for vaccinia virus). All infectivity data is reported as average of three biological replicates (independent infections followed by titering) plus and minus the standard deviation.

## Western blotting for protein expression

The following commercially available primary antibodies were used: mouse M2 anti-Flag (Sigma-Aldrich, St. Louis, MO; F1804), rabbit anti-hCMTR1 (Sigma-Aldrich; HPA029979), mouse anti-PGK1 (Invitrogen, Carlsbad, CA; 459250), and rat anti-tubulin (Millipore, Temecula, CA; CBL270). Goat anti-mouse, anti-rat and anti-rabbit HRP-conjugated secondary antibodies were from Santa Cruz Biotechnology (Dallas, TX).

For analysis of protein expression from stably transduced BSC40 cells, $\sim$200000 cells were harvested and lysed by boiling in 2x SDS sample buffer. For analysis of protein expression from yeast, a single yeast colony was picked from a freshly streaked plate of transformed yeast and grown overnight in liquid selective media containing 2% raffinose (SC -his/-leu RAF). Cultures were diluted in SC -his/-leu RAF to 0.5 $OD_{600}$ and grown 3 more hours at which point galactose was added to a final concentration of 2%. After 90 min, cultures were spun down and frozen. Cell pellets were resuspended in 2x SDS sample buffer containing protease inhibitors and bead beaten for 30 s. All cell lysates (yeast and mammalian) were run on 4–12% Bis-Tris gels (Invitrogen) and transferred onto nitrocellulose membrane. Blocking buffer and antibody dilution buffer for hCMTR1 blots was 5% bovine serum albumin in phosphate buffered saline (PBS) with 0.1% Tween-20. Blocking buffer and antibody dilution buffer for all other blots was 5% nonfat dried milk in PBS with 0.1% Tween-20.

## Acknowledgements

We thank members of the Malik and Geballe labs for helpful discussions and advice. We especially thank Michael Emerman, Kevin Forsberg, Patrick Mitchell, and Janet Young for comments on the manuscript and Stephanie Child for help with vaccinia virus preparations. This study was supported by a postdoctoral fellowship from the Cancer Research Institute (MDD), National Institutes of Health NIH grant under award numbers RO1AI027762 (APG) and R21AI099936 (APG/HSM), and funds from the Howard Hughes Medical Institute (HSM). HSM is an HHMI Investigator. The content is solely the responsibility of the authors and does not necessarily represent the official views of the National Institutes of Health or other funding agencies.

## Additional information

### Funding

| Funder | Grant reference number | Author |
| --- | --- | --- |
| National Institute of Allergy and Infectious Diseases | RO1AI027762 | Adam P Geballe |
| Howard Hughes Medical Institute | Investigator award | Matthew D Daugherty Harmit S Malik |
| Cancer Research Institute | Postdoctoral fellowship | Matthew D Daugherty |
| National Institute of Allergy and Infectious Diseases | R21AI099936 | Adam P Geballe Harmit S Malik |

The funders had no role in study design, data collection and interpretation, or the decision to submit the work for publication.

### Author contributions

MDD, Conception and design, Acquisition of data, Analysis and interpretation of data, Drafting or revising the article; AMS, Acquisition of data, Drafting or revising the article; APG, HSM, Conception and design, Analysis and interpretation of data, Drafting or revising the article

Author ORCIDs
Matthew D Daugherty, http://orcid.org/0000-0002-4879-9603
Harmit S Malik, http://orcid.org/0000-0001-6005-0016

## Additional files

### Supplementary files

• Supplementary file 1. IFIT sequences and accession numbers used for phylogenetic studies.

• Supplementary file 2. Oligonucleotides used for cloning and manipulation of IFIT genes.

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
