## [Decision Letter]

Thank you for submitting your article "Evolution-guided functional analyses reveal diverse antiviral specificities encoded by *IFIT1* genes in mammals" for consideration by *eLife*. Your article has been reviewed by two peer reviewers, one of whom, Wesley Sundquist, is a member of our Board of Reviewing Editors, and the evaluation has been overseen by Diethard Tautz as the Senior Editor.

The reviewers have discussed the reviews with one another and the Reviewing Editor has drafted this decision to help you prepare a revised submission.

General assessment:

The authors describe the evolutionary history of the IFIT genes, and convincingly demonstrate that they fall into two separate families, IFIT1 and IFIT1B. Their analyses reveal that the two protein families recognize different features on mRNAs, with IFIT1B proteins recognizing mRNAs that lack 2'O-methylation at their 5' caps. The analyses also: 1) sort out the true relationships between murine and human *IFIT1* genes, demonstrating that human *IFIT1* and murine *IFIT1B* (new names) are paralogs, not orthologs, 2) demonstrate that the C-terminal domains define the IFIT1- vs. IFIT1B-type target specificity, and 3) a number of different primate lineages have lost either *IFIT1* and *IFIT1B* genes, or undergone *IFIT/IFIT1B* gene conversion/recombination events.

Overall, this paper is very clearly written, the experimental and computational analyses are convincing, a clever new yeast assay for restriction activity is presented (as are complementary viral restriction data), and the field has been clarified. As the authors acknowledge, it is a significant limitation that we still don't understand how IFIT1 proteins distinguish non-self from self. The authors also have to argue that human IFIT1B is probably a non-functional protein, which is quite plausible but is based on negative data. Those caveats aside, however, this is a significant and technically strong paper.

This study also exemplifies the complexities of recapitulating evolution of genes with dynamic evolutionary trajectories such as anti-pathogen genes that encounter multiple, ever-shifting selective pressures across evolutionary timeframes. It provides a very strong illustration of how simple phylogenetic reconstructions can be misleading, and the utility of incorporating other information including synteny and locus structure as characters in such analyses – thus, this study could be seen as a model for reanalyzing other genes in this category (e.g., the myriad ISG effectors) and possibly revisiting other innate immune gene families that have been analyzed in the past.

Issues for the authors' consideration

In the Discussion section, paragraph four, the authors make the statement that most viruses have evolved ways of circumventing IFIT1B. This seems to fall into the category of inferring past function from current utility – one can't really say that IFIT1B selected for these strategies and it is alternatively possible that the various mechanisms that allow evasion of IFIT1B evolved before IFIT1B. There could be, for example, many other antiviral strategies in nature that center on viral mRNA and translation, and until the mechanism of IFIT1B is understood, this claim cannot be made.

In Discussion, the authors clearly prefer a model in which IFIT1-like proteins discriminate "self from non-self" (I might prefer "viral" from "cellular"), but they could use a clearer outline of the arguments for their preference over a model in which it is simply a tightly regulated but otherwise global repressor of translation? – for example, via a combination of IFN induction and active repression, a second messenger, synergy with other IFIT proteins, or some other mechanism not present in yeast, IFIT1 may act globally under conditions when a cell has been breeched by a virus. Can this be ruled out, or at least can arguments be made that this is less likely?

The study of how large gene families evolve is not new, and the authors should make sure to cite some of the prior work in this vein. Papers that helped establish the role of gene conversion in evolution of multi-gene families are relevant here, for example. Similarly, in the first paragraph of the discussion, it would be nice to cite other published examples of innate immune gene families that have similarly complex evolutionary histories (in addition to citing the single review article written by the authors).

The use of hashtags to indicate bootstrap support is unusual, as is the cutoff of 75% (typically, 80 is used, although values of 75% probably won't change the conclusions of the paper). There seems to be sufficient space in the figure that values should be given next to the corresponding nodes.

---

## [Author Response]

The authors describe the evolutionary history of the IFIT genes, and convincingly demonstrate that they fall into two separate families, IFIT1 and IFIT1B. Their analyses reveal that the two protein families recognize different features on mRNAs, with IFIT1B proteins recognizing mRNAs that lack 2'O-methylation at their 5' caps. The analyses also: 1) sort out the true relationships between murine and human IFIT1 genes, demonstrating that human IFIT1 and murine IFIT1B (new names) are paralogs, not orthologs, 2) demonstrate that the C-terminal domains define the IFIT1- vs. IFIT1B-type target specificity, and 3) a number of different primate lineages have lost either IFIT1 and IFIT1B genes, or undergone IFIT/IFIT1B gene conversion/recombination events.

Overall, this paper is very clearly written, the experimental and computational analyses are convincing, a clever new yeast assay for restriction activity is presented (as are complementary viral restriction data), and the field has been clarified. As the authors acknowledge, it is a significant limitation that we still don't understand how IFIT1 proteins distinguish non-self from self. The authors also have to argue that human IFIT1B is probably a non-functional protein, which is quite plausible but is based on negative data. Those caveats aside, however, this is a significant and technically strong paper.

This study also exemplifies the complexities of recapitulating evolution of genes with dynamic evolutionary trajectories such as anti-pathogen genes that encounter multiple, ever-shifting selective pressures across evolutionary timeframes. It provides a very strong illustration of how simple phylogenetic reconstructions can be misleading, and the utility of incorporating other information including synteny and locus structure as characters in such analyses – thus, this study could be seen as a model for reanalyzing other genes in this category (e.g., the myriad ISG effectors) and possibly revisiting other innate immune gene families that have been analyzed in the past.

Issues for the authors' consideration

In the Discussion section, paragraph four, the authors make the statement that most viruses have evolved ways of circumventing IFIT1B. This seems to fall into the category of inferring past function from current utility – one can't really say that IFIT1B selected for these strategies and it is alternatively possible that the various mechanisms that allow evasion of IFIT1B evolved before IFIT1B. There could be, for example, many other antiviral strategies in nature that center on viral mRNA and translation, and until the mechanism of IFIT1B is understood, this claim cannot be made.

We agree with the reviewers and amended our earlier phrasing, which implied that viruses evolved these means to specifically counteract IFIT1B. Instead we now state that “However, the general antiviral effectiveness of IFIT1B proteins is limited due to the fact that every mammalian virus family, except alphaviruses, has evolved a way to produce cap1-mRNAs.”

*In Discussion, the authors clearly prefer a model in which IFIT1-like proteins discriminate "self from non-self" (I might prefer "viral" from "cellular"), but they could use a clearer outline of the arguments for their preference over a model in which it is simply a tightly regulated but otherwise global repressor of translation? – for example, via a combination of IFN induction and active repression, a second messenger, synergy with other IFIT proteins, or some other mechanism not present in yeast, IFIT1 may act globally under conditions when a cell has been breeched by a virus. Can this be ruled out, or at least can arguments be made that this is less likely?*

Although in this instance the reviewer is correct that “self vs non-self” is similar to “host vs viral” this is how the discrimination is referred to in the field in general. However, we now lay out a clear rationale for why we parsimoniously favor the model that IFIT1 recognizes a novel mRNA pattern to inhibit translation: “Our study does not elucidate the molecular means by which human IFIT1 inhibits viral replication. While several possibilities exist for how IFIT1-mediated translational repression might be controlled, we favor the possibility that IFIT1, like its paralog IFIT1B, distinguishes a 'self versus non-self' pattern on mRNA to selectively inhibit viral replication. Supporting this model is the fact that in addition to IFIT1B, IFIT2 and IFIT5 have been reported to have sequence- or structure-specific binding to RNA (Habjan et al., 2013; Katibah et al., 2013; Katibah et al., 2014; Yang et al., 2012), suggesting that numerous IFITs may restrict viral replication through recognition of distinct viral RNA patterns. Although the presumed molecular pattern that IFIT1 recognizes remains unknown, our data suggest that mammalian hosts, as well as vaccinia virus, possess a 'self' molecular pattern to prevent IFIT1-mediated inhibition whereas yeast and VSV display a 'non-self' molecular pattern that cannot block IFIT action.”

The study of how large gene families evolve is not new, and the authors should make sure to cite some of the prior work in this vein. Papers that helped establish the role of gene conversion in evolution of multi-gene families are relevant here, for example. Similarly, in the first paragraph of the discussion, it would be nice to cite other published examples of innate immune gene families that have similarly complex evolutionary histories (in addition to citing the single review article written by the authors).

We agree with this advice by the reviewers and now cite seven studies or reviews showing gene conversion in multigene families (Results section) and eight studies or reviews of innate immune gene families with complex evolutionary histories (Discussion section) in our revision.

The use of hashtags to indicate bootstrap support is unusual, as is the cutoff of 75% (typically, 80 is used, although values of 75% probably won't change the conclusions of the paper). There seems to be sufficient space in the figure that values should be given next to the corresponding nodes.

We agree with the reviewers comment and have now added actual bootstrap values when they exceed 80% in all the phylogenetic figures.